# Silk Fibroin-Induced Gadolinium-Functionalized Gold Nanoparticles for MR/CT Dual-Modal Imaging-Guided Photothermal Therapy

**DOI:** 10.3390/jfb13030087

**Published:** 2022-06-22

**Authors:** Chuanxue Yang, Tianxiao Mei, Qingge Fu, Yifan Zhang, Yang Liu, Ran Cui, Gang Li, Yibin Wang, Jianguo Huang, Junqiang Jia, Bo Chen, Yihui Hu

**Affiliations:** 1Institute for Regenerative Medicine, Jian Hospital & Shanghai East Hospital, School of Life Sciences and Technology & School of Medicine, Tongji University, Shanghai 200092, China; yangchuanxueff@163.com (C.Y.); meitianxiao@tongji.edu.cn (T.M.); y.i.zhang@hotmail.com (Y.Z.); liuyang0711@126.com (Y.L.); cuiangus@tongji.edu.cn (R.C.); hjg2015@163.com (J.H.); 2Department of Hepatopancreatobiliary Surgery, Jian Hospital & Shanghai East Hospital, Jian 343000, China; 3Department of Orthopedic Trauma, Changhai Hospital, Naval Medical University, Shanghai 200433, China; fqg7821@163.com; 4Department of Radiology, Shanghai East Hospital, School of Medicine, Tongji University, Shanghai 200092, China; ligangace@163.com (G.L.); yalding1136@163.com (Y.W.); 5School of Grain Science and Technology, Jiangsu University of Science and Technology, Zhenjiang 212100, China

**Keywords:** silk fibroin, AuNPs, biomimetic synthesis, MR/CT imaging, PTT

## Abstract

The development of multifunction nanoplatforms integrating accurate diagnosis and efficient therapy is of great significance for the precise treatment of tumors. Gold nanoparticles (AuNPs) possessing hallmark features of computed tomography (CT) imaging and photothermal conversion capability hold great potential in tumor theranostics. In this study, taking the advantages of outstanding biocompatibility, interesting anti-inflammatory and immunomodulatory properties, and abundant amino acid residues of silk fibroin (SF), a multifunctional Gd-hybridized AuNP nanoplatform was constructed using SF as a stabilizer and reductant via a facile one-pot biomimetic method, denoted as Gd:AuNPs@SF. The obtained Gd:AuNPs@SF possessed fascinating biocompatibility and excellent photothermal conversion efficiency. Functionalized with Gd, Gd:AuNPs@SF exhibited super tumor-contrasted imaging performance in magnetic resonance (MR) and CT imaging modalities. Moreover, Gd:AuNPs@SF, with strong NIR absorbance, demonstrated that it could effectively kill tumor cells in vitro, and was also proved to successfully ablate tumor tissues through MR/CT imaging-guided photothermal therapy (PTT) without systemic toxicity in Pan02 xenograft C57BL/6 mouse models. We successfully synthesized Gd:AuNPs@SF for MR/CT dual-mode imaging-guided PTT via a facile one-pot biomimetic method, and this biomimetic strategy can also be used for the construction of other multifunction nanoplatforms, which is promising for precise tumor theranostics.

## 1. Introduction

Nowadays, multifunctional nanomaterials that combine diagnostic imaging and therapeutic activity are receiving increasing attention as novel theranostic agents for precision medicine, especially for cancer therapy [1,2]. To date, a variety of nanoplatforms have been established to achieve accurate imaging for tumor diagnosis, and thus for guiding therapy [3,4,5,6,7]. MR and CT are currently the most common clinical diagnostic techniques; nevertheless, conventional single-modal contrast agents cannot meet the needs of complex diagnostics. MR imaging can provide clear soft tissue imaging that is non-invasive without radiation, yet the sensitivity is unsatisfactory; while CT imaging has a fast acquisition speed and high time resolution, it is not clear for soft tissue imaging [8,9,10]. Thus, MR/CT dual-modal imaging is able to confer more informative and accurate imaging diagnoses for tracking tumors in vivo with satisfactory spatial and temporal resolution [11]. Therefore, it would be highly worthwhile to develop a theranostic system that combines MR/CT dual-modal imaging diagnosis and efficient therapy for the accurate elimination of cancer.

AuNPs have been applied extensively in tumor theranostics [12,13]. To date, AuNPs have been synthesized using various methods including Turkevich, Brust, biomimetic synthesis, etc. [14,15]. These AuNPs have played different roles, such as imaging contrast agents for precise diagnosis, PPT agents for cancer therapy, and in novel theranostic strategies [16,17,18]. Notably, biomolecule-mediated, especially protein-induced, biomimetic synthesis possesses the unique advantages of green friendliness, versatility, designability, and biocompatibility, and thus it has attracted tremendous interest in the nanofabrication of biomedical applications. For example, recent work by Yang et al. presented the biomimetic synthesis of Gd:CuS@BSA with high biocompatibility for photoacoustic (PA)/MR-guided tumor-targeted PTT [19]. The team of Xing et al. also constructed collagen–gold hybrid hydrogels through the biomineralization of collagen, for tumor PTT and photodynamic therapy (PDT) [20].

Silk fibroin (SF) is a natural protein extracted from the cocoon of the silkworm (*Bombyx mori*) and has been widely applied in tissue engineering and biomimetic synthesis because of its outstanding biocompatibility, low immunogenicity, marvelous anti-inflammatory properties and immuneregulation effect, and biodegradability [21,22,23,24,25,26]. SF contains abundant amino acid residues, which confer reaction sites and linking groups for biosynthesis and modification [27,28,29,30]. In particular, the molecular structural framework and excellent water stability of SF make it suitable as a green reducing and stabilizing agent for fabricating nanostructures. A previous study reported SF@MnO_2_ nanoparticles constructed by biomineralization crystallization, and further loaded photodynamic agent (indocyanine green) and chemotherapeutic (doxorubicin) for MR/fluorescence imaging-guided PTT/PDT/chemotherapy [31]. SF can also be used as a template to synthesize AuPt nanozymes that respond to the tumor microenvironment for enhancing tumor catalytic therapy [32]. We summarize the conventional and the emerging protein-mediated biomineralization synthesis methods of AuNPs, as well as the synthesis of SF nanoparticles, in Appendix A. Surprisingly, SF-mediated Gd-hybridized AuNPs for MR/CT dual-modal imaging-guided precise therapy has yet to be explored.

Herein, we developed a one-pot reducing strategy for the biomimetic synthesis of Gd-functionalized AuNPs (named, Gd:AuNPs@SF) mediated by SF as a reducer and stabilizer (Figure 1). The obtained Gd:AuNPs@SF exhibits several advantages: (1) high biocompatibility and stability, (2) reliable MR/CT bimodal imaging diagnosis, and (3) excellent photothermal conversion for PTT. Furthermore, favoring MR/CT dual-modal imaging-guided PTT for precise pancreatic cancer therapy was demonstrated in vitro Pan02 cells and in vivo Pan02-bearing pancreatic cancer mice. Therefore, this work explores a facile one-pot biomimetic method to construct multifunctional AuNPs theranostic agents, allowing for precision nanomedicine.

## 2. Materials and Methods

### 2.1. Materials

Silk fibroin was provided by the School of Biotechnology, Jiangsu University of Science and Technology. Chloroauric acid tetrahydrate (HAuCl_4_·4H_2_O), gadolinium(III) chloride hexahydrate (GdCl_3_·6H_2_O), chlorpromazine hydrochloride, genistein, and NaOH were obtained from Shanghai Mackin Biochemical Co., Ltd. Wortmannin was purchased from Aladdin (Shanghai, China). Dulbecco’s Modified Eagle’s Medium (DMEM), fetal bovine serum (FBS), and Dulbecco’s phosphate buffered saline (PBS) were purchased from Biological Industries. Acridine Orange/Propidium Iodide (AO/PI) stain was obtained from Nexcelom Bioscience. Cell counting kit-8 (CCK-8) was ordered from Yeasen Biotech Co., Ltd., (Shanghai, China). Fluorescein isothiocyanate (FITC) was supplied by Shanghai Mackin Biochemical Co., Ltd., (Shanghai, China). All aqueous solutions used in experiments were prepared with deionized water (18.2 MΩ·cm resistivity at 25 °C).

### 2.2. Instruments

Transmission electron microscopy (TEM, JEM-2100, JEOL, Tokyo, Japan) and field emission scanning electron microscopy (SEM, JSM-7800F, JEOL, Tokyo, Japan) were performed to observe the morphology and size. The X-ray photoelectron spectroscopy (XPS) measurements were performed using a PHI-5000 CESCA system (PerkinElmer, Waltham, MA, USA) with radiation from an Al Kα (200.0 eV) X-ray source. X-ray energy spectrum analysis (EDS) was used to test the elemental composition. Fourier transform infrared spectroscopy (FTIR) characterization was performed using a Fourier Infrared Spectrophotometer (TENSOR 27, Bruker, Karlsruhe, Baden-Württemberg, German). Hydrodynamic diameter and Zeta potential were measured on a dynamic light scattering particle size analyzer (Zetasizer Nano-ZS90, Malvern, Great Malvern, Worcestershire, UK). The photothermal effect was tested on an 808 nm laser (Beijing Energylaser Tech. Ltd., Beijing, China). Temperature changes and thermal images were monitored and taken using an IR thermal camera (DALI Technology, Hangzhou, China). The absorbance at 450 nm of the CCK-8 test and UV-vis absorption spectra were measured with a multifunction microplate reader (Infinite M200 Pro, TECAN, Männedor, Switzerland). Fluorescence photographs were obtained on a fluorescence microscope (Leica Microsystems, GER, Wezler, Hesse, German). The Gd and Au concentration was measured by ICP-OES (Agilent 725 ICP-OES, Agilent Technologies Co. Ltd., Palo Alto, CA, USA). MR imaging was applied to the clinical MR imaging system (GE Discovery 750W 3.0 T, General electric medical system Co., Ltd., Fairfield, CT, USA), and CT imaging was performed using a micro-CT system (SkySCAN 1276, Bruker, Karlsruhe, Baden- Württemberg, German). The cellular uptake efficiency was analyzed using flow cytometry (CytoFLEX, Beckman coulter, Brea, CA, USA).

### 2.3. Synthesis of Gd:AuNPs@SF

The synthesis of Gd:AuNPs@SF was adapted from the BSA-mediated biomimetic synthesis of Au NPs with some minor changes [33]. Briefly, HAuCl_4_·4H_2_O solution (2 mM, 5 mL) was added into the SF aqueous solution (3.125 mg/mL, 2 mL), and NaOH solution (1 M) was used to adjust the pH to 9–10; then, GdCl_3_·6H_2_O solution (29.9 mM, 80 μL) was slowly added to the mixed solution. The mixture solution was kept stirring at room temperature for 6–8 h to obtain a dark red solution. The solution was then centrifuged (12,000 rpm, 10 min) and washed three times with H_2_O to obtain pure Gd:AuNPs@SF for further use. After lyophilization, a black powder was obtained for characterizations.

The synthesis of Gd:AuNPs@SF/FITC was simply, FITC (2 mg/mL, dissolved in absolute ethanol), added into 1 mg/mL Gd:AuNPs@SF under stirring, reacted overnight in the dark, and then centrifuged (12,000 rpm, 10 min) to remove unreacted FITC to obtained Gd:AuNPs@SF/FITC.

### 2.4. Stability Assays

To determine the colloidal stability, Gd:AuNPs@SF were dispersed in water, DMEM, 1 × PBS (pH 7.4) and stored for 7 days, the samples were taken at different time points (2 h, 6 h, 12 h, 24 h, 48 h and 7 days) to determine the hydrodynamic diameter, Zeta potential, and UV-vis absorbance to monitor the stability of nanoparticles.

### 2.5. Photothermal Performance of Gd:AuNPs@SF

First, 200 μL Gd:AuNPs@SF (250 μg/mL) aqueous solution and water was put into a dish, and vertically irradiated for 6 min with different laser power (0.5, 1.0, 2.0, 2.5, 3.0 W/cm^2^) at 808 nm near-infrared (a spot with a diameter of 6.39 mm). During this period, the temperature was recorded every 30 s with an infrared imager. Temperature changes were recorded, the heating curve was drawn, and photothermal conversion efficiency was calculated. Through 5 cycles of turning the laser on and off, the cyclic heating curve of the material was obtained.

### 2.6. Hemolysis Test

The hemolysis of the Gd:AuNPs@SF was characterized by co-incubating it with human red blood cells according to the method reported in the literature [19]. Briefly, 1 mL blood samples were diluted by 2 mL of PBS, and then centrifuged at 12,000 rpm for 10 min to obtain red blood cells, washed four times with PBS, and diluted with 10 mL of PBS. The diluted red blood cells were mixed with 1 mL of Gd:AuNPs@SF at different concentrations (0, 10, 25, 50, 100, 150, 200, 250 μg/mL), while 1 mL of water was used as a positive control and 1 mL of PBS as a negative control. After incubating for 2 h at 37 °C, the solutions were centrifuged at 12,000 rpm for 10 min, and the supernatant was added into a 96-well plate to measure the absorbance at 570 nm with a microplate reader for calculating the hemolysis rate. Hemolysis ratio (%) = (A(sample) − A(PBS))/(mean value of A(H_2_O)−A(PBS))×100%.

### 2.7. Cell Culture and Toxicity Analysis

Mouse pancreatic cells (Pan02, 3111C0001CCC000446) were purchased from the National Infrastructure of Cell Line Resource (China). Human Umbilical Vein Endothelial Cells (HUVEC) were purchased from Cell Bank, Shanghai Institute of Life Science (China Academy of Science). Pan02 and HUVEC cells were incubated in DMEM containing 10% FBS and 1% antibiotics (penicillin-streptomycin) at 37 °C in a humidified atmosphere containing 5% CO_2_.

When Pan02 and HUVEC cells had grown to 80–90% of the culture flask, they were seeded into a 96-well plate (5 × 10^3^ cells/well) and incubated for 24 h; then, they were replaced with a fresh culture medium containing gradient concentrations of Gd:AuNPs@SF (0, 10, 25, 50, 100, 150, 200, 250 μg/mL), and after 24 or 48 h incubation, washed twice with PBS. Then, DMEM medium containing 10% CCK-8 was added, followed by incubation for 1–2 h at 37 °C and 5% CO_2_, and then measurement of the absorbance at 450 nm with a microplate reader.

The photothermal cytotoxicity of Gd:AuNPs@SF was measured by CCK-8 assay and AO/PI staining. Pan02 cells (5 × 10^3^ cells/well) were plated on 96-well plates (two groups), cultured in a constant temperature incubator at 37 °C and 5% CO_2_ for 24 h. Group 1 was replaced with a fresh culture medium containing gradient concentrations of Gd:AuNPs@SF (0, 10, 25, 50, 100, 150, 200, 250 μg/mL), and directly exposed to the laser (808 nm, 3.0 W/cm^2^, 6 min, a spot with a diameter of 6.39 mm), then washed twice with PBS and incubated DMEM medium with 10% CCK-8 for 1–2 h at 37 °C and 5% CO_2_. Finally, the absorbance at 450 nm was measured with a microplate reader. Group 2 was used for AO/PI staining. After treatment with PBS (200 μL), only Gd:AuNPs@SF (250 μg/mL, 200 μL), only laser (808 nm, 3.0 W/cm^2^, 6 min), and Gd:AuNPs@SF (250 μg/mL, 200 μL) + laser (808 nm, 3.0 W/cm^2^, 6 min), respectively, were stained by AO/PI staining to observe cell viability.

### 2.8. Cellular Uptake

Pan02 cells were seeded in a 24-well plate (1 × 10^5^ cells/well) overnight. Cells were pre-incubated with chlorpromazine (10 μg/mL), genistein (200 μg/mL), wortmannin (100 ng/mL), or DMEM medium supplemented with 10% FBS for 1 h. Then, the medium was removed, and washed twice with PBS. Gd:AuNPs@SF/FITC of 250 μg/mL was added for another 3 h incubation. Finally, the cells were collected by centrifugation (1000 rpm, 5 min), and resuspended in 200 μL of PBS for flow cytometry.

### 2.9. In Vitro Imaging

In vitro MR imaging was performed on the basis of MR imaging signal intensity of Gd:AuNPs@SF with different Gd^3+^ concentrations (0.04, 0.08, 0.12, 0.16, 0.20 mM).The detailed imaging parameters of MR were set as follows: T1 weighted sequence, spin echo, FOV reading = 9.0 mm, FOV phase = 1.00 mm, slice = 6, slice thickness = 1.0 mm, slice gap = 0.2 mm. The CT signal intensity of Gd:AuNPs@SF with different Au^3+^ concentrations (0.05, 0.10, 0.20, 0.30, 0.40 mM) to examine CT imaging capabilities. The imaging parameters were as follows: 80 kV, 200 μA, filter: AI 0.5 mm, exposure: 137 ms. The software provided by the instrument manufacturer was used to collect images.

### 2.10. Animal Model Establishment and In Vivo Imaging

All the animal experimental protocols were approved by the Institutional Animal Care and Use Committee of Tongji University (SYXK(hu)2020-0002). Mouse models of pancreatic cancer were used for animal experiments. The process of establishing an animal model was as follows: Pan02 cells (3 × 10^6^ cells) were suspended in 100 μL of PBS solution and injected subcutaneously into the hind legs of C57 female mice to establish a mice model of pancreatic cancer. When the tumors had grown to 50–100 mm^3^, they were used for in vivo imaging and photothermal therapy. Tumor-bearing mice were used for in vivo MR/CT imaging. Tumor-bearing mice were injected with 200 μL Gd:AuNPs@SF (1.5 mg/mL) or PBS via tail vein. Clinical imaging system was used to obtain MRI signals before and 2 h, 4 h, 6 h, 12 h, and 24 h after injection. CT imaging was performed on the Micro-CT system before and 2 h and 6 h after injection.

### 2.11. Biodistribution

First, 200 μL of 1.5 mg/mL Gd:AuNPs@SF was injected into the tail vein of Pan02 tumor-bearing mice (*n* = 3). The mice were sacrificed at 6 h after injection, and then the major organs (heart, liver, spleen, kidney, lung) and tumor were collected, weighed and digested using aqua regia (6 mL, HCl/HNO_3_ = 3:1) overnight. The concentration of Au in the tissues was quantitatively detected by ICP-OES.

### 2.12. In Vivo Photothermal Therapy

The tumor-bearing mice were randomly divided into four groups (*n* = 3), (1) Gd:AuNPs@SF (1.5 mg/mL, 200 μL) + laser (808 nm, 0.8 W/cm^2^, 6 min); (2) only Gd:AuNPs@SF (1.5 mg/mL, 200 μL); (3) only laser (808 nm, 0.8 W/cm^2^, 6 min); (4) PBS (200 μL). During laser irradiation, an infrared imager was used to monitor the temperature at the tumor site. The body weight and tumor volume of the mice were monitored every two days.

### 2.13. Histological Analysis

Mice were sacrificed after 22 days of treatment, and the main organs (heart, liver, spleen, kidney, lung) and tumors were collected and fixed with 4% paraformaldehyde; then, tissues were paraffin embedded, sectioned, hematoxylin-eosin stained, observed, and photographed by microscope.

### 2.14. Statistical Analysis

Data was treated by Graphpad Prism 7.0 software. Statistical analysis used mean ± standard deviation (SD) and was conducted by ANOVA. *p* values were obtained using the 3-sample *t*-test. A *p*-value less than 0.05 (* *p* < 0.05, *n* = 3) was considered statistically significant.

## 3. Results and Discussion

### 3.1. Synthesis and Characterization of Gd:AuNPs@SF

Multifunctional Gd:AuNPs@SF were successfully synthesized as a theranostic nanoplatform via the biomimetic one-pot method (Figure 1), where SF acted as the reducing agent and the stabilizer. Because of the intense affinity of carboxyl groups to metal ions, Gd^3+^ and Au^3+^ species were absorbed on SF. Au^3+^ was reduced to Au^+^ by the amino group, which was further reduced to Au by tyrosine’s cresol groups due to their strong electron-donating ability [34,35]. During the synthesis, the mixture solution of SF, HAuCl_4_·4H_2_O, and GdCl_3_·6H_2_O gradually turned from pale yellow to dark red when the NaOH solution was added to adjust the pH to 9–10, suggesting that the tyrosine of SF exhibited reducing properties under alkaline conditions and thereby reduced Au ions into AuNPs (Figure 1A). The influence of the concentration of SF (0.78125 mg/mL, 3.125 mg/mL, 25 mg/mL) on the AuNPs’ synthesis was investigated. According to TEM imaging and UV-vis spectra (Appendix A), Gd:AuNP@SF displayed a uniform sphere with suitable size and possessed maximum absorption when the concentration of SF at 3.125 mg/mL, which was set as the optimal concentration.

The morphology of the optimized Gd:AuNPs@SF measured by TEM and SEM was a regular sphere with an average diameter of 53.88 ± 1.81 nm, (Figure 1B,C). Its hydrodynamic diameter detected by dynamic light scattering (DLS) was 162.0 ± 42.69 nm (Figure 1D), which was larger than that detected by TEM because of the hydration layer of SF. The Zeta potential on the surface of Gd:AuNPs@SF was −26.6 ± 1.33 mV owing to the abundant carboxyl groups of SF (Figure 1E). As shown in Figure 1F, Gd:AuNPs@SF had a characteristic absorption of AuNPs at 535 nm and displayed a broad absorption in the near-infrared region (600–900 nm) for PTT. The EDS mapping showed that Au, Gd, and N were distributed in the structures of Gd:AuNPs@SF, which indicated that the elements Au, Gd, and SF were the main constituents (Figure 1G). The composition of Gd:AuNPs@SF was further determined by EDS (Figure 1H) and XPS (Figure 1I and Appendix A), confirming the existence of Au, Gd, C, N, and O elements. Additionally, the XPS spectra of Au (4f), Gd (4d), N (1s), and C (1s) were analyzed using XPS Peak 4.0 software (Appendix A). It can be claimed that peaks at 88.0 eV, 84.5 eV, 87.2 eV and 83.5 eV in the Au (4f) spectrum can be allocated to Au^3+^, Au^+^, Au^0^, respectively, elucidating the formation of AuNPs. The peaks at 148.0 eV and 143.0 eV in the Gd (4d) spectrum were assigned to Gd_2_O_3_ [36]. This is slightly different from what has previously been reported, which usually claims the presence of both Gd(OH)_3_ and Gd_2_O_3_ [19,37,38]. This might be due to the introduction of SF, which is able to stabilize the entire reaction system and prevent the production of the insoluble Gd(OH)_3_. FTIR spectra were used to examine the conformation of SF and Gd:AuNPs@SF. As shown in Figure 1J, the peaks at 3187 cm^−1^ of Gd:AuNPs@SF and 3382 cm^−1^ of SF were amide N-H contraction vibration absorption peaks, revealing the presence of SF and the generation of the unsaturated C-H bond. SF showed amide characteristic peaks at 1610 cm^−1^ (amide I), 1522 cm^−1^ (amide II), 1243 cm^−1^ (amide III). However, the amide I, II, III peaks in Gd:AuNPs@SF moved to 1760 cm^−1^, 1647 cm^−1^, 1208 cm^−1^, respectively, and the amide peaks (I, II) shifted to longer wavenumbers, which may suggest an increase of random coil structure according to a previous study [39]. These pieces of evidence suggested that the binding of Gd and Au to SF could change the conformation of SF. The interaction between hydrogen bonds and amide groups of SF and metal ions might contribute to the shift of the peaks. In general, the required materials of Gd:AuNPs@SF were successfully synthesized.

In addition, the colloidal stability of Gd:AuNPs@SF was studied by observing dispersed Gd:AuNPs@SF in water, PBS, and DMEM. Monitored the UV-vis absorption and particle size distribution within 7 days. As presented in Appendix A, the UV absorption spectrum does not change significantly within 7 days, and the hydrodynamic diameter fluctuates around 160 nm, keeping relatively stable. The digital photo illustrates that no significant aggregation of Gd:AuNPs@SF was observed within 7 days. The above results indicate that Gd:AuNPs@SF were stable in these solution systems.

### 3.2. Photothermal Effect and Stability of Gd:AuNPs@SF

Firstly, the effect of laser power on photothermal actuation was investigated. As shown in Figure 2A, Gd:AuNPs@SF (250 μg/mL) were irradiated with an NIR laser at 808 nm with different power densities (0.5, 1.0, 2.0, 2.5, 3.0 W/cm^2^). After 6 min of irradiation, the temperature increased by 2.4, 2.9, 8.1, 21.4, and 27.1 °C, respectively, and exhibited a laser power-density-dependent pattern. Under 808 nm laser irradiation (3.0 W/cm^2^, 6 min), the temperature of water only increased by 12.8 °C, much lower than Gd:AuNPs@SF (27.1 °C), indicating that the Gd:AuNPs@SF had good photothermal conversion capabilities (Figure 2B). Then, different concentrations of Gd:AuNPs@SF (0, 25, 100, 150, 250 μg/mL) were exposed to 3.0 W/cm^2^ NIR laser, demonstrating that the temperature increase was concentration dependent (Figure 2C,D). In addition, infrared thermal images were used to monitor the photothermal effect of Gd:AuNPs@SF, demonstrating that the temperature of Gd:AuNPs@SF could reach up to 54.7 °C (3.0 W/cm^2^, 6 min), but the temperature of water only reached 25.4 °C (3.0 W/cm^2^, 6 min) (Figure 2G). According to Roper’s report and the Lambert–Beer law, the photothermal conversion efficiency of Gd:AuNPs@SF was 21.63% (Figure 2E and Appendix A) [40]. As shown in Figure 2F, after five cycles of 808 nm laser irradiation, the maximum temperature of Gd:AuNPs@SF still reached 56 °C, demonstrating excellent photothermal stability. Therefore, the robust hyperthermia and high photothermal stability of Gd:AuNPs@SF allow for effective ablation of cancer cells.

### 3.3. Cellular Safety and Photothermal Toxicity

The safety and biocompatibility of Gd:AuNPs@SF were assessed on Pan02 cells, HUVEC cells, and red blood cells. The viability of Pan02 cells (Figure 3A) and HUVEC cells (Appendix A) maintained levels of up to 75% even when incubated for 48 h at a high concentration of Gd:AuNPs@SF (250 μg/mL). Similarly, when the concentration of Gd:AuNPs@SF was up to 250 μg/mL, the hemolysis rate did not exceed 2% (Figure 3B). These results indicate that Gd:AuNPs@SF possess high biosafety and excellent biocompatibility, which is favorable for in vivo administration.

Moreover, Gd:AuNPs@SF showed significant photothermal toxicity on Pan02 cells in vitro. As shown in Figure 3C, the viability of Pan02 cells (less than 11% of Pan02 cells survived) decreased significantly after NIR irradiation (3.0 W/cm^2^, 6 min) at different concentrations of Gd:AuNPs@SF, while there was almost no apoptosis without NIR irradiation. AO/PI staining also verified that nearly all of the Pan02 cells died after being exposed to NIR irradiation, which was consistent with the CCK-8 assay (Figure 3D).

To reveal the cellular uptake mechanism of Gd:AuNPs@SF, we used flow cytometry to detect the intracellular FITC content with Gd:AuNPs@SF/FITC. As shown in Appendix A, FITC labeling has no significant effect on the particle size of Gd:AuNPs@SF. Chlorpromazine inhibits clathrin-mediated endocytosis, genistein inhibits caveolae-mediated endocytosis, and wortmannin inhibits macro-pinocytosis. The inhibitors were incubated with Pan02 cells for 1 h and then Gd:AuNPs@SF/FITC was added for another 3 h incubation. As shown in Appendix A, chlorpromazine displayed an obvious inhibitory effect on cell uptake, illustrating clathrin-mediated endocytosis might be the main pathway of Gd:AuNPs@SF uptake.

### 3.4. Ex Vivo and In Vivo MR/CT Imaging of Gd:AuNPs@SF

To explore the MR and CT imaging properties of Gd:AuNPs@SF in vitro, MR imaging and CT imaging were carried out for different concentrations of Gd^3+^ ions (0–0.20 mM) and Au^3+^ ions (0–20 mM) in Gd:AuNPs@SF, respectively. As shown in Figure 4A and Appendix A, Gd:AuNPs@SF showed a stronger T1 signal than Gd-DTPA at the same concentration of Gd^3+^ ions, implying that Gd:AuNPs@SF was an excellent MR contrast agent. Analogously, the CT imaging characteristic was the same as MR imaging (Figure 4B).

Furthermore, the tumor enrichment of Gd:AuNPs@SF was investigated by MR and CT imaging in Pan02 tumor-bearing mice. Following injection of Gd:AuNPs@SF, the T1 signal at the tumor sites increased 1.4-fold, 3.96-fold, 4.52-fold, 3.77-fold, 1.67-fold at 2, 4, 6, 12 and 24 h, respectively (Figure 4C,D). These results indicate that the MR imaging signal of the tumor site was the strongest at 6 h after intravenous injection, suggesting the plateau of Gd:AuNPs@SF enrichment in the tumors. Subsequently, CT imaging was also performed by the Micro-CT imaging system. According to the results from MR imaging, we obtained CT imaging before injection and 2 h and 6 h after injection of Gd:AuNPs@SF. Remarkably, the CT signal at the tumor site also was significantly enhanced 6 h after injection (Figure 4E), which was in line with the MR imaging. The MR/CT imaging results verified that Gd:AuNPs@SF have the capability of MR/CT dual-modal imaging, and the content reached the maximum at the tumor site after 6 h injection, which could guide PTT.

To investigate the biodistribution of Gd:AuNPs@SF, after 6 h injection of 200 μL of Gd:AuNPs@SF solution (1.5 mg/mL), the tumors and the main organs (heart, liver, spleen, lung, kidney) of the mice were harvested and analyzed with respect to Au concentration by ICP-OES (Appendix A). As shown in Figure 4F of the distribution diagram, although most of Gd:AuNPs@SF were distributed in the spleen and lung, the cumulative efficiency at the tumor site could still be up to 10% ID/g. In this work, Gd:AuNPs@SF mainly relied on enhanced permeability and retention effect (EPR) to achieve accumulation at tumor sites without specific tumor targeting. Therefore, Gd:AuNPs@SF were also distributed in other tissues. As reported, antibody-coupled nanoparticles showed stronger binding force and internalization ability for tumor cells [41,42]. Antibody-coupled nanoparticles also promoted the death of cancer cells with over-expression of antigen [43]. In follow-up research, antibodies or ligands could also be coupled with the gold nanoparticles to increase tumor targeting and further improve the therapeutic effect of tumors.

### 3.5. In Vivo Photothermal Therapy and Histology of Gd:AuNPs@SF

To evaluate whether Gd:AuNPs@SF could generate effective heat in vivo for PTT, the temperature evolution at tumor sites of Pan02-bearing mice without or with Gd:AuNPs@SF treatment was recorded under 808 nm laser irradiation. As shown in Figure 5A,B, the temperature increased from 30 °C to 47.9 °C in the Gd:AuNPs@SF group with laser irradiation; however, the temperature of the control group (PBS + laser) only increased from 30 °C to 39.4 °C. Compared with the control group, the temperature of Gd:AuNPs@SF increased by 17.9 °C, almost 1.9-fold that of the PBS group (9.4 °C), which indicated that the Gd:AuNPs@SF had good photothermal conversion ability in vivo and was sufficient for the ablation of tumor cells.

Further, photothermal treatment in vivo was performed on tumor-bearing mice transplanted subcutaneously with Pan02 cells. Tumor-bearing mice were randomly divided into four groups (*n* = 3): Gd:AuNPs@SF + laser, Gd:AuNPs@SF only, laser only, PBS. As shown in Figure 5C–E, the tumor volume of the Gd:AuNPs@SF + laser group quickly faded after PTT, while the tumors in other groups (PBS, single laser, and single Gd:AuNPs@SF group) continued to grow, indicating that the heat generated by the Gd:AuNPs@SF under 808 nm laser irradiation for 6 min was sufficient to ablate the tumor tissue. In addition, there was no significant change in body weight for any of the groups, and it remained relatively stable (Appendix A), showing that the photothermal treatment had no systemic toxicity. Hematoxylin–eosin (H&E) staining of tumor tissue after 22 days of treatment is shown in Figure 5F; PBS, single laser, and single Gd:AuNPs@SF treatment did not result in significant cell necrosis, while the tumor with the Gd:AuNPs@SF + laser treatment completely faded. Meanwhile, the main organs of the mice were collected for H&E staining (Appendix A), illustrating that the major organs were intact after PTT. These results demonstrate that the Gd:AuNPs@SF + laser treatment program can not only successfully ablate tumor tissues, but also, that there is no marked biological toxicity to other tissues and organs.

The blood vessel structures of tumors are incomplete, making their heat dissipation ability poor, resulting in heat accumulation inside. Meanwhile, the heat tolerance of tumor cells is lower than that of normal cells. The lethal temperature of tumor cells is 42.5–43 °C, while normal cells can tolerate a high temperature of 47 °C, which is an important reason why PTT can effectively kill cancer cells [44]. In this study, the temperature of the tumor site in the Gd:AuNPs@SF + laser group was 47.9 °C, which is much higher than the lethal temperature for tumor cells, and thus eliminated the tumor cells. As we look forward, the tumor volume in mice also showed that Gd:AuNPs@SF + laser treatment effectively ablated tumor tissues. In addition, the inherent anti-tumor activity of AuNPs may also play an important role. AuNPs with radiosensitization effects in cells could induce the production of ROS, trigger oxidative stress, lead to mitochondrial dysfunction, and perturb normal activities of cells [45]. Studies have also demonstrated that AuNPs can selectively accumulate in the mitochondria of tumor cells, reducing mitochondrial membrane potential and increasing oxidative stress, and eventually causing tumor cell apoptosis [46].

## 4. Conclusions

In summary, we proposed a facile one-pot biomimetic method to prepare Gd-functionalized AuNPs, named Gd:AuNPs@SF, which constitute a promising MR/CT contrast agent and photothermal agent for precise tumor theranostics. The obtained Gd:AuNPs@SF possessed high biocompatibility and safety, favorable photothermal effect, and reliable MR/CT dual-modal imaging capabilities. Under the guidance of MR/CT bimodal imaging, accurate and effective PTT could be achieved. Therefore, the SF biomimetic system not only provides a novel multifunctional platform for precise diagnosis and treatment of tumors, but also possesses the potential to be functionalized with other contrast agents or drugs.

## Data Availability

The data presented in this study are available in the article and Appendix A.

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
