# Peer review of "Silk Fibroin-Induced Gadolinium-Functionalized Gold Nanoparticles for MR/CT Dual-Modal Imaging-Guided Photothermal Therapy"

_jfb, 2022, doi:10.3390/jfb13030087_

Round 1

Reviewer 1 Report

In their manuscript entitled “Silk fibroin-induced gadolinium-functionalized gold nanoparticle for MR/CT dual-modal imaging-guided photothermal therapy”, authors presented an interesting preclinical study using their theranostic Au-Gd nanoparticles. The manuscript is well written, presents original results but these should be contextualized in the more general field of nanomedicine in oncology. You will find my comments below :

1.     Introduction line 39-40:A variety of nanoplatforms have been established to achieve accurate imaging for tumor diagnosis, and thus for guiding therapy [3-5]”. A very important reference in the field (doi.org/10.1088/1361-6560/ab9159) made by a group of experts is missing.

2.     Materials & Methods: Could you add the size of the irradiation field to the beam characteristics?

3.     Figure 2. Why are there no error bars on your experimental data? Have the results been repeated? If so, how many independent replicates are there?

4.     Figure 2G: Can you specify degrees Celsius on the right column?

5.     Result section 3.2: Can you extract from your data (figure 2D), a value of temperature variation per unit of concentration?

6.     Result section 3.3 : You modify your nanoparticle by adding a fluorescent probe (FITC). To what extent could the addition of this probe modify the internalization of the nanoparticle? Do you have an idea of the size differences of the 2 nano-objects (with and without FITC)?

7.     Overall comment: Statistics are missing on your whole data set. Example: figure 3, Which cell viability data are significantly different from the control?

8.     Figure 4: Could you add a relative value of signal per unit of Gadolinium concentration?

9.     Figure 4F: Do you get the same biodistribution results if you quantify Gadolinium and not gold? This would ensure that the Gd and Au do not separate from each other during IV injection for any reason.

10. Section 3.4. Seeing your biodistribution results (% ID/g tissue), I think it would be interesting to discuss these in relation to the results obtained by injection of active nanoparticles (example : gold nanoparticles coupled to antibodies).

11. Section 3.4. it would be interesting to extract a Gd/Au concentration in the tumor in order to correlate your in vivo results (Fig. 5A and B) with the in vitro ones (Fig. 2). I refer you to my comment n°5.

12. Figure 5C: Could you calculate a Coefficient of drug interaction (CDI) to determine if the observed effect is additive or synergistic?

13.  There is a lack of a general mechanistically oriented discussion. You seem to only believe that heat is generated by light - nanoparticle interaction and that this heat is responsible for cell death. If we take a step back from that and look at the latest developments in nanomedicine, it has been shown that the presence of gold and other metals in cancer cells induces a series of biological dysfunctions whose amplitude correlates with the sensitizing effect of the irradiation. It would be interesting to see if you observe the same effects of enzymatic inhibition of thioredoxin reductase, which would greatly orient the mechanistic discussion that is missing. If you do not have the opportunity to perform these additional experiments, a mechanistic discussion of the potential impact of nanoparticles on these biological alterations would still be important.

Interesting work. Congratulation !

Reviewer 2 Report

Increasing the therapeutic efficacy in oncological diseases is an urgent problem of modern oncology. A new multifunctional platform is developed by the authors in this article. It allows you to simultaneously visualize the location of the tumor and conduct thermotherapy. Interest for silk fibroin and gadolinium as a platform for the development of new therapeutic agents already exists (Gold nanomaterials functionalised with gadolinium chelates and their application in multimodal imaging and therapy //Chem. Commun., 2020,56, 4037-4046, Study of magnetic silk fibroin nanoparticles for massage-like transdermal drug delivery//International Journal of Nanomedicine, 2015, 10(1):4639-51).

But there are limitations to this approach, this is a low depth of penetration of the laser beam into living tissues. The limitation is superficial variants of the tumor, although laser fibers can solve this problem.

Another significant point is the allergenic and immunogenic properties of the developed drug based on silk fibroin. It is a foreign protein for warm-blooded animals. Although the obtained Gd:AuNPs@SF hold high biocompatibility and safety for cells in vitro test, there is no data on animal response in vivo test.

Nevertheless, the article is of interest to readers and may be accepted for publication.

Round 2

Reviewer 1 Report

I thank the authors for this very interesting revised version. Here are my last comments:

* Regarding your answer 3 : Could you clearly add in the legend for each experiment the number of replicates to which these results refer (n= 1 or n=2)?

* Regarding your answer 6 : Could you add this interesting graph in supplemental data ?

* Regarding your answer 7 : Could you add the information regarding statistics in the figure caption ? At the moment, I do not see any information in the caption or in a "statistic section" in Mat & Met that explain to the reader what  "****" means (statistical threshold considered for p-value ? Statistical test used ? )

* Regarding your answer 9 : Could you add this table in supplemental (after translating it in english) ? What is the ratio between Au and Gd in the nano-object ? Is it possible to explain this content difference by the Au/Gd molar ratio in the nanoparticle ? 

* Regarding your answer 10 : Please add this discussion in the main text. 

* Figure 5A : Can you specify degrees Celsius on the right column?

* Regarding your answer 13 : Your discussion is really interesting but you do not integrate it in your main text. Please add this discussion to your main text. 

Round 3

Reviewer 1 Report

Thank you for your answers